# STEALTHY IMITATION: REWARD-GUIDED ENVIRONMENT-FREE POLICY STEALING

## ABSTRACT

Deep reinforcement learning policies, which are integral to modern control systems, represent valuable intellectual property. The development of these policies demands considerable resources, such as domain expertise, simulation fidelity, and real-world validation. These policies are potentially vulnerable to model stealing attacks, which aim to replicate their functionality using only black-box access. In this paper, we propose Stealthy Imitation, the first attack designed to steal policies without access to the environment or knowledge of the input range. This setup has not been considered by previous model stealing methods. Lacking access to the victim's input states distribution, Stealthy Imitation fits a reward model that allows to approximate it. We show that the victim policy is harder to imitate when the distribution of the attack queries matches that of the victim. We evaluate our approach across diverse, high-dimensional control tasks and consistently outperform prior data-free approaches adapted for policy stealing. Lastly, we propose a countermeasure that significantly diminishes the effectiveness of the attack. The implementation of Stealthy Imitation will be publicly available and open-source.

## 1 INTRODUCTION

Neural networks trained with reinforcement learning (RL), known as deep RL policies, are increasingly employed in control systems due to the exceptional performance and automation capabilities. Developing a reliable deep RL policy requires substantial resources, including expertise in training, precise simulation, and real-world testing; the resulting policy becomes important intellectual property. However, neural network models are vulnerable to stealing attacks (Tramèr et al., 2016; Orekondy et al., 2019b; Truong et al., 2021) that attempt to copy the functionality of the model via black-box query access. The risks posed by such attacks are manifold, including unauthorized model usage, exposure of sensitive information, and further attacks based on the leaked model.

Model theft typically consists of two steps. First, a transfer dataset is created by querying the victim model with publicly available data (Orekondy et al., 2019b), random noise (Tramèr et al., 2016), or samples synthesized by a neural network (Truong et al., 2021), and recording the model predictions as pseudo-labels. The latter two methods fall under the category of data-free model stealing. After this querying phase, the attackers train their own model via supervised learning, treating the pseudo-labels as ground truth for their samples.

Control systems, such as industrial automation, remotely controlled drones or robots, pose additional challenges for model stealing. A policy perceives states and rewards (also known as the environment), based on which it decides the next action to take. In this context, the attacker can potentially send queries to the system, but does not have access to the environment. Data-free stealing attacks hold the promise of environment-free policy stealing. While existing data-free attacks have proven effective in the image domain, they operate under the assumption that the attacker knows the valid input range. For instance, valid image pixels are assumed to be in the range of $[0, 255]$. However, such prior knowledge is difficult to acquire in control systems or other applications, due to the distinct semantics and scales of components within the measured state. As a consequence, policy stealing becomes more difficult.

To address this challenge, we introduce Stealthy Imitation (SI), the first environment-free policy stealing attack. Our method solves the two fundamental difficulties of this task: (i) the necessity of accurately estimating the input range and distribution of the states visited by the victim policy, and

(ii) the identification of a metric that allows the attacker to evaluate the estimated distribution, and thus its own performance in stealing the policy. These insights collectively enable a more robust and efficient policy stealing attack. Notably, the derived distribution remains applicable even when the victim updates their policy without altering the training distribution, offering potential savings in query budget for subsequent attacks.

**Contributions.** (i) We introduce a more general and realistic threat model adapted to control systems, where the attacker lacks access to the environment and to the valid input ranges of the policy. (ii) We propose Stealthy Imitation (SI), the first reward-guided environment-free policy stealing method under minimal assumptions. We show our attack to be effective on multiple control tasks. (iii) We introduce the first proxy metric to measure the quality of the estimated distribution. We empirically and statistically validate its correlation with the divergence between the estimated distribution and the actual state distribution of the victim policy. (iv) We develop a defense that is able to counter the proposed attack, thus offering a practical solution for practitioners.

## 2 RELATED WORK

**Knowledge distillation.** Knowledge distillation (KD) was initially designed for model compression, aiming to approximate a large neural network (commonly referred to as the teacher model) with a more compact model (the student model). This facilitates deployment on hardware with limited computational capabilities (Ba & Caruana, 2014; Hinton et al., 2015). Unlike our work, which adopts an adversarial view, KD typically presumes access to the teacher model's original training dataset, enabling the student model to learn under the same data distribution. When the dataset is large or sensitive, some methods opt for surrogate datasets (Lopes et al., 2017). Others eliminate the need for it by employing data generators in data-free KD approaches (Fang et al., 2019; Micaelli & Storkey, 2019). These methods often assume white-box access to the teacher model for backpropagation, which is a major difference with our setup.

**Model stealing.** Model stealing focuses on adversarial techniques for the black-box extraction of a victim model (equivalent to the teacher model in KD) (Tramèr et al., 2016; Orekondy et al., 2019b). The attacker, who aims to create a surrogate model (analogous to the student in KD), lacks access to the original training dataset of the victim model. Most existing methods explore data-free stealing, drawing inspiration from data-free knowledge distillation, but lacking the means to use the victim model to train a data generator. These techniques estimate the gradient of the victim model for training their generator and encourage query exploration by synthesizing samples that maximize the disagreement between victim and attacker model (Sanyal et al., 2022; Beetham et al., 2022; Truong et al., 2021). While much work has been conducted in image-based domains, limited research exists on model stealing in the context of reinforcement learning (Chen et al., 2021; Behzadan & Hsu, 2019). Our approach sidesteps the need for environment access and specific knowledge of the RL algorithm employed by the victim. Existing defenses primarily focus on detecting stealing attacks (Juuti et al., 2019; Kesarwani et al., 2018) or perturbing model predictions (Tramèr et al., 2016; Orekondy et al., 2019a). Our proposed defense falls in the latter category: the policy perturbs its outputs when the query falls outside the valid input range.

**Imitation learning.** Imitation learning aims to train agents to emulate human or expert model behavior. Within this domain, there are two main methodologies. The first is behavioral cloning (BC), which treats policy learning as a supervised learning problem, focusing on state-action pairs derived from expert trajectories (Pomerleau, 1991). The second is inverse reinforcement learning, which seeks to discover a cost function that renders the expert's actions optimal (Russell, 1998; Ng et al., 2000). Another method of interest is generative adversarial imitation learning (GAIL), which utilizes adversarial training to match the imitating agent's policy to that of the expert. Notably, GAIL achieves this alignment using collected data and does not need further access to the environment (Ho & Ermon, 2016). Our work deviates from these imitation learning approaches, as we do not require access to the interaction data between the expert policy, i.e., the victim for us, and its environment.

## 3 THREAT MODEL

In this section, we formalize the threat model for black-box policy stealing in the context of deep RL policies used in control systems. First, we introduce preliminary concepts and notations. Then, we formalize the victim's policy. Finally, we outline the attacker's knowledge and the relevance of this threat model to real world attacks.

**Notations.** In the context of deep RL, a policy or agent, is denoted by $\pi$ with accepting state $\boldsymbol{s}$, and predicting an action $\boldsymbol{a}$, such that $\boldsymbol{a} = \pi(\boldsymbol{s})$. A trajectory $\tau \sim \pi$ consists of a sequence of states and actions collected from the interaction between policy and environment. We represent the initial state distribution as $\rho_0$, and the environment's state transition function as $f$, such that $\boldsymbol{s}_{t+1} = f(\boldsymbol{s}_t, \boldsymbol{a}_t)$. The return, or cumulative reward, for a trajectory is represented as $R(\tau)$, while $S$ is the distribution of states visited by the deployed policy.

**Victim policy.** We consider a victim operating a deep RL policy, $\pi_v$, trained to optimize a particular control objective with accepting one time step state $\boldsymbol{s} \in \mathbb{R}^n$ and predicting the action $\boldsymbol{a}^* \in \mathbb{R}^k$ within the range of $[-1, 1]$. The environment is fully observable by the victim policy. The performance of the policy is quantified using the expected return $\mathbb{E}_{\tau_v \sim \pi_v}[R(\tau_v)]$ in the deployed setting. $S_v$ represents the distribution of states visited by $\pi_v$.

**Goal and knowledge of the attacker.** We take on the role of the attacker, with the goal of training a surrogate policy $\pi_a$ to replicate the functionality of the victim policy $\pi_v$ to achieve similar (average) return in the environment. The attacker possesses black-box access to $\pi_v$ by querying states and obtaining actions as responses. However, the attacker lacks knowledge on several key aspects: (i) the internal architecture and RL training algorithm of $\pi_v$, (ii) the environment setup, including the initial state distribution $\rho_0$, the state-transition function $f$, and the reward function $R$, (iii) the semantics associated with the input and output spaces, (iv) the range of the inputs, as well as the state distribution $S_v$, and (v) the confidence score of all possible actions from the victim policy. This lack of knowledge makes policy stealing particularly challenging.

**Real-world relevance.** Our threat model emerges from the critical need to identify vulnerabilities in systems employing deep RL policies, for example remotely accessible control systems. It highlights the significance of environment-free scenarios, where attackers, lacking environment insights, aim to replicate functionalities through black-box policy queries, a situation where obtaining data through direct interaction with the system is unfeasible. Successful policy theft in such scenarios compromises intellectual property, privacy, and augments the risk of subsequent security attacks.

## 4 APPROACH: STEALTHY IMITATION

This section introduces the details of Stealthy Imitation. The method overview in Section 4.1 is followed by an explanation of each of its components in Section 4.2. Section 4.3 shows how to use the estimated distributions from prior steps to steal the target policy. Lastly, we propose a defense that can make the attacker's goal more difficult to reach.

### 4.1 METHOD OVERVIEW

We introduce Stealthy Imitation as attacker that steals a policy without access to the environment or to the valid input range. To achieve their goal, the attacker aims to optimize the surrogate policy $\pi_a$ to minimize the expected return difference between their own policy $\pi_a$ and that of the victim $\pi_v$ in the environment:

$$\underset{\pi_a}{\arg\min} \left| \underset{\tau_a \sim \pi_a}{\mathbb{E}} [R(\tau_a)] - \underset{\tau_v \sim \pi_v}{\mathbb{E}} [R(\tau_v)] \right| \tag{1}$$

However, the attacker does not have access to the environment or the reward function. Instead, they can minimize the action difference between their policy and that of the victim on an estimated state distribution $S_a$ using a loss function $\mathcal{L}$ as a proxy for the reward:

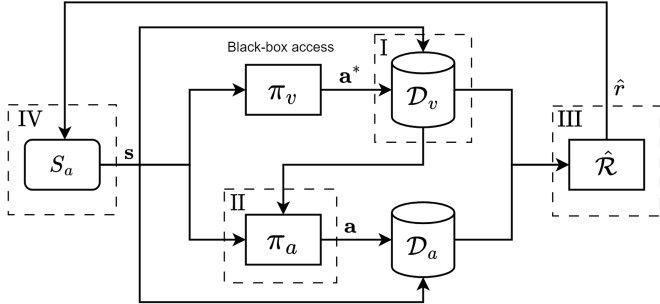

Figure 1: Overview of Stealthy Imitation that iteratively refines the estimated state distribution $S_a$.

$$\arg\min_{\pi_a} \mathbb{E}_{\boldsymbol{s} \sim S_a} [\mathcal{L}(\pi_v(\boldsymbol{s}), \pi_a(\boldsymbol{s}))] \tag{2}$$

The attacker's goal is thus to find both the victim policy and the appropriate distribution of states. The Stealthy Imitation objective encourages exploration by maximizing the disagreement between the victim and attacker models:

$$\overset{\textbf{II}}{\arg\min_{\pi_a}} \overset{\textbf{III \& IV}}{\arg\max_{S_a}} \mathbb{E}_{\boldsymbol{s} \sim S_a} \left[ \mathcal{L}(\pi_a(\boldsymbol{s}), \overset{\textbf{I}}{\pi_v(\boldsymbol{s})}) \right]. \tag{3}$$

The core of Stealthy Imitation consists of four main steps repeated iteratively until the attacker query budget $B$ is consumed: (I) **transfer dataset construction** by querying the victim policy with states sampled from the estimated distribution $S_a$; (II) training the attacker policy $\pi_a$ via **behavioral cloning** to mimic the victim policy on the transfer dataset; (III) **reward model training** $\hat{R}$ to discriminate the behaviours of the victim and current attacker policy, and (IV) **reward-guided distribution refinement** to closer match the victim's state distribution using the proxy reward score on each query state. Once the attacker's budget is exhausted, we train $\pi_a$ from scratch only on the best estimated distribution with the help of a distribution evaluator. The approach overview is depicted in Figure 1. We detail each step in the following.

## 4.2 State Distribution Estimation

**I. Transfer dataset construction.** As the attacker has no knowledge of the state distribution of the victim $S_v$, we choose a multivariate normal distribution $\mathcal{N}(\boldsymbol{\mu}, \boldsymbol{\sigma}^2)$ with a diagonal covariance matrix as estimate of the attacker distribution $S_a$ ($S_a(\boldsymbol{s}; \boldsymbol{\mu}, \boldsymbol{\sigma})$ from here on). States $\boldsymbol{s}$ are sampled from this distribution and passed to the victim policy to obtain corresponding actions $\boldsymbol{a}^*$. The transfer dataset $\mathcal{D}_v$ described below is split into training and validation for use in the subsequent method steps:

$$\mathcal{D}_v = \{(\boldsymbol{s}, \boldsymbol{a}^*)\}, \quad \text{where} \quad \boldsymbol{s} \sim S_a(\boldsymbol{s}; \boldsymbol{\mu}, \boldsymbol{\sigma}), \quad \text{and} \quad \boldsymbol{a}^* = \pi_v(\boldsymbol{s}). \tag{4}$$

Lacking prior knowledge, $S_a$ is initialized with $\boldsymbol{\mu} = \mathbf{0}_n$ and $\boldsymbol{\sigma} = \mathbf{1}_n$. In each iteration, we use a dynamic query budget by multiplying a base budget $b_v$ with the average of $\boldsymbol{\sigma}$. This ensures sufficient learning in mimicking the actions of the victim policy, especially when the estimated $\boldsymbol{\sigma}$ is large, thereby stabilizing the refinement process.

**II. Behavioral cloning.** We follow the conventional step in model stealing to mimic the victim policy's behavior using the training split of the transfer dataset $\mathcal{D}_v$. To this end, we employ behavioral cloning using Huber loss (Huber, 1964), known for its resilience to outliers:

$$\mathcal{L}_b(\pi_a(\boldsymbol{s}), \boldsymbol{a}^*) = \begin{cases} 0.5(\pi_a(\boldsymbol{s}) - \boldsymbol{a}^*)^2 & \text{if } |\pi_a(\boldsymbol{s}) - \boldsymbol{a}^*| < 1, \\ |\pi_a(\boldsymbol{s}) - \boldsymbol{a}^*| - 0.5 & \text{otherwise.} \end{cases} \tag{5}$$

**III. Reward model training.** To evaluate the difficulty of imitation from the state-action pairs from $\pi_a$ and $\pi_v$, we introduce a reward model. Our approach is motivated by the hypothesis that the complexity of victim's responses increases when $S_a$ approximates $S_v$, thus it becomes more challenging for the attacker policy $\pi_a$ to accurately imitate the victim's behaviors. This hypothesis is supported empirically by the results in Section 5.3. We adopt a discriminative classifier $\hat{\mathcal{R}}$, inspired by GAIL (Ho & Ermon, 2016). The role of $\hat{\mathcal{R}}$ is to distinguish between state-action pairs generated by the victim and attacker policy. A more effective distinction suggests that the attacker's policy is more challenging to imitate accurately. To this end, we construct a dataset $\mathcal{D}_a$ using actions $\mathbf{a}$ generated by $\pi_a(\mathbf{s})$ after BC, and train a reward model $\hat{\mathcal{R}}$ by minimizing the loss function $\mathcal{L}_r$:

$$\mathcal{L}_r(\boldsymbol{s}, \boldsymbol{a}) = \mathbb{E}_{(\boldsymbol{s}, \boldsymbol{a}) \sim \mathcal{D}_a} [-\log(\hat{\mathcal{R}}(\boldsymbol{s}, \boldsymbol{a}))] + \mathbb{E}_{(\boldsymbol{s}, \boldsymbol{a}^*) \sim \mathcal{D}_v} [-\log(1 - \hat{\mathcal{R}}(\boldsymbol{s}, \boldsymbol{a}^*))]. \tag{6}$$

**IV. Reward-guided distribution refinement.** We use the trained reward model from the previous step $\hat{\mathcal{R}}$ to generate proxy reward values $\hat{r}(\boldsymbol{s}, \boldsymbol{a}) = -\log(\hat{\mathcal{R}}(\boldsymbol{s}, \boldsymbol{a}))$ for each state-action pair. A high reward value $\hat{r}(\boldsymbol{s}, \boldsymbol{a}^*)$ indicates that the attacker policy fails to effectively mimic the victim, suggesting that the state has higher probability in $S_v$. These reward values serve as weights for the corresponding samples $\boldsymbol{s}$, which we use to recompute the parameters $\boldsymbol{\mu}'$ and $\boldsymbol{\sigma}'$ of the distribution for the next iteration, as follows:

$$\begin{aligned} \boldsymbol{\mu}' &= \frac{\sum_{(\boldsymbol{s}, \boldsymbol{a}^*) \in \mathcal{D}_v} \hat{r}(\boldsymbol{s}, \boldsymbol{a}^*) \cdot \boldsymbol{s}}{\sum_{(\boldsymbol{s}, \boldsymbol{a}^*) \in \mathcal{D}_v} \hat{r}(\boldsymbol{s}, \boldsymbol{a}^*)}, \\ \boldsymbol{\sigma}'^2 &= \frac{\sum_{(\boldsymbol{s}, \boldsymbol{a}^*) \in \mathcal{D}_v} \hat{r}(\boldsymbol{s}, \boldsymbol{a}^*) \cdot (\boldsymbol{s} - \boldsymbol{\mu}')^2}{\sum_{(\boldsymbol{s}, \boldsymbol{a}^*) \in \mathcal{D}_v} \hat{r}(\boldsymbol{s}, \boldsymbol{a}^*)}. \end{aligned} \tag{7}$$

### 4.3 POLICY STEALING ON THE ESTIMATED DISTRIBUTION

Since the attacker has no knowledge of the victim states' distribution $S_v$, we introduce a model $\pi_e$, which we term distribution evaluator. This model helps assess the closeness between the attacker and victim distributions $S_a$ and $S_v$. $\pi_e$ is trained via behavioral cloning and is reinitialized in each iteration to ensure its validation loss $\bar{\mathcal{L}}_b$ measures only the error of the current estimated distribution. Based on our hypothesis, a higher loss $\bar{\mathcal{L}}_b$ is indicative of $S_a$ closely mirroring $S_v$. We only use $b_v$ samples of the transfer dataset $\mathcal{D}_v$ to train $\pi_e$ instead of $b_v \times \bar{\boldsymbol{\sigma}}$. This ensures it is only affected by the distribution divergence without the influence of training data size. Once the attacker budget is exhausted, i.e., the algorithm is done iterating over steps I-IV, the parameters $\tilde{\boldsymbol{\mu}}$ and $\tilde{\boldsymbol{\sigma}}$ from the iteration that yielded the highest loss value are used to create an optimized transfer dataset using the remaining reserved query budget $B_r$. Finally, $\pi_a$ is subsequently retrained from scratch via BC using this optimized dataset. Algorithm 1 outlines the complete method; all the functions used are defined in Appendix A.

### 4.4 STEALTHY IMITATION COUNTERMEASURE

Although this work focuses on the attacker's perspective, we also propose an effective defense against Stealthy Imitation. We argue that ignoring queries outside the valid range is not advisable for the victim, as it would leak information about the valid range itself. The idea is to leverage the victim's exclusive knowledge of the correct input range; the defender can respond with random actions to invalid queries. This approach serves to obfuscate the attacker's efforts to estimate the input range. This defense does not degrade the utility of the victim policy, as it still provides correct answers to valid queries.

## 5 EXPERIMENTS

This section presents our empirical results for Stealthy Imitation. We discuss the experimental setup (Section 5.1), followed by a comparison of our proposed method to baselines (Section 5.2) and analyses and ablation studies (Section 5.3). Finally, we show the defense performance in Section 5.4.

---

**Algorithm 1** Stealthy Imitation

---

**Require:** Victim policy $\pi_v$ (blackbox access), total budget $B$, reserved budget $B_r$, base query budgets $b_v$ and
 $b_a$ for victim and attacker victims respectively in each iteration
**Ensure:** Trained attacker policy $\pi_a$
 1: Initialize attacker policy $\pi_a$, distribution evaluator $\pi_e$, reward model $\hat{\mathcal{R}}$, $\boldsymbol{\mu} \leftarrow \mathbf{0}_n$, $\boldsymbol{\sigma} \leftarrow \mathbf{I}_n$
 2: Initialize proxy metric $\tilde{\mathcal{L}} \leftarrow -\infty$, consumed budget $B_c \leftarrow 0$, and to be consumed budget $b_c \leftarrow b_v$
 3: **while** $B_c + b_c < B - B_r$ **do**
 4:     $\mathcal{D}_v \leftarrow \text{QueryAction}(\pi_v, \boldsymbol{\mu}, \boldsymbol{\sigma}, b_c)$                                    ▷ I. Transfer dataset construction
 5:     $\bar{\mathcal{L}}_b \leftarrow \text{DistributionEvaluate}(\mathcal{D}_v, \pi_e, b_v)$                                    ▷ Section 4.3
 6:     **if** $\bar{\mathcal{L}}_b > \tilde{\mathcal{L}}$ **then**
 7:         $\tilde{\mathcal{D}}, \tilde{\mathcal{L}}, \tilde{\boldsymbol{\mu}}, \tilde{\boldsymbol{\sigma}} \leftarrow \mathcal{D}_v, \bar{\mathcal{L}}_b, \boldsymbol{\mu}, \boldsymbol{\sigma}$
 8:     **end if**
 9:     $\pi_a \leftarrow \text{BehavioralCloning}(\mathcal{D}_v, \pi_a, b_v \cdot \bar{\boldsymbol{\sigma}})$                                    ▷ II. Behavioral cloning
10:     $\mathcal{D}_a \leftarrow \text{QueryAction}(\pi_a, \boldsymbol{\mu}, \boldsymbol{\sigma}, b_a)$
11:     $\hat{\mathcal{R}} \leftarrow \text{TrainReward}(\mathcal{D}_a, \mathcal{D}_v, \hat{\mathcal{R}}, b_v \cdot \bar{\boldsymbol{\sigma}})$                                    ▷ III. Reward model training
12:     $\boldsymbol{\mu}, \boldsymbol{\sigma} \leftarrow \text{DistRefine}(\mathcal{D}_v, \hat{\mathcal{R}}, b_v \cdot \bar{\boldsymbol{\sigma}})$                                    ▷ IV. Reward-guided distribution refinement
13:     $B_c \leftarrow B_c + b_c$
14:     $b_c \leftarrow \max(b_v, b_v \cdot \bar{\boldsymbol{\sigma}})$
15: **end while**
16: $\tilde{\mathcal{D}} \leftarrow \tilde{\mathcal{D}} \cup \text{QueryAction}(\pi_v, \tilde{\boldsymbol{\mu}}, \tilde{\boldsymbol{\sigma}}, B - B_c)$
17: $\pi_a \leftarrow \text{BehavioralCloning}(\tilde{\mathcal{D}}, \pi_a, |\tilde{\mathcal{D}}|)$ with reinitialized $\pi_a$
18: **return** $\pi_a$

---

## 5.1 EXPERIMENTAL SETUP

**Victim policies.** We demonstrate our method on three continuous control tasks from Mujoco (Todorov et al., 2012): Hopper, Walker2D, and HalfCheetah. The victim policy is trained using soft actor-critic (SAC) (Haarnoja et al., 2018). The victim architecture is a three-layer fully-connected networks (256 hidden units, ReLU activation). The models output a normal distribution from which actions are sampled. These sampled actions are then constrained to the range [-1,1] using tanh. After training, the prediction action given a query state is determined only by the mean of this output distribution. See Appendix B for a complete description of all the tasks and performance of the victim policies. In addition, Stealthy Imitation is also preliminarily validated in a robot arm setting, as described in Appendix G.

**Attacker policies.** Similar to Papernot et al. (2016); Orekondy et al. (2019b;a), we employ the architecture of $\pi_v$ for $\pi_a$, while omitting the prediction of the standard deviation and incorporating tanh activation. Our choice of architecture does not significantly influence the refinement of $S_a$ (see Appendix C), although it does introduce greater variance in the cumulative reward. This phenomenon is attributed to compounding errors, a known issue in imitation learning (Syed & Schapire, 2010; Ross et al., 2011; Xu et al., 2020), where minor training deviations can amplify errors. We set the reserved training budget $B_r = 10^6$ and the base query budget $b_v = 10^5$. Both $\pi_a$ and $\pi_e$ share the same architecture and are trained for one epoch per iteration. We use the Adam optimizer (Kingma & Ba, 2015) with a learning rate of $\eta = 10^{-3}$ and batch size of 1024. The final training employs early stopping with a patience of 20 epochs for 2000 total epochs. The reward model $\hat{\mathcal{R}}$ is a two-layer fully-connected network (256 hidden neurons, tanh and sigmoid activations). $\hat{\mathcal{R}}$ is trained with a learning rate of 0.001 for 100 steps. Prior to training, we apply a heuristic pruning process to $\mathcal{D}_v$. Specifically, we remove any state-action pairs $(\boldsymbol{s}, \boldsymbol{a})$ where any component of $\boldsymbol{a}$ equals $\pm 1$, corresponding to the maximum and minimal action values. This pruning step is motivated by the preference for stability in control systems: actions of large magnitude are typically avoided. This further assists the reward model in correctly identifying the victim policy's state-action pattern.

**Baseline attacks.** Since our method is the first policy stealing without environment access or prior input range knowledge, we compare it against two approaches: (i) Random: transfer datasets are based on three normal distributions with varying scales, namely $\mathcal{N}(\mathbf{0}_n, \mathbf{1}_n^2)$, $\mathcal{N}(\mathbf{0}_n, \mathbf{10}_n^2)$, and $\mathcal{N}(\mathbf{0}_n, \mathbf{100}_n^2)$; the attacker policy $\pi_a$ is trained using BC; (ii) data-free model extraction (DFME): we adapt the generator-based DFME (Truong et al., 2021) from image classification to control tasks.

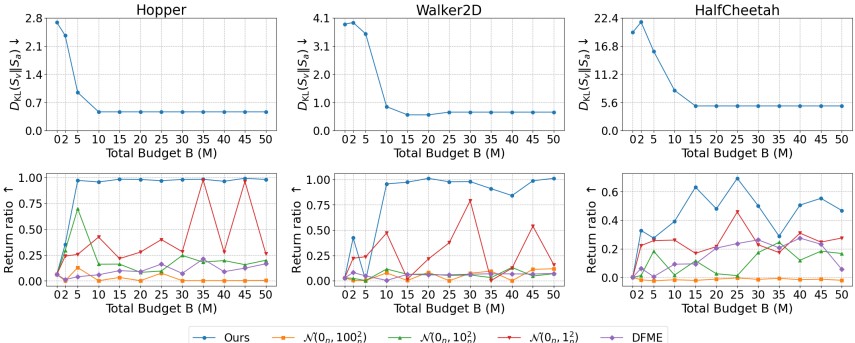

Figure 2: Distribution estimation capacity measured by $D_{\mathrm{KL}}(S_v\|S_a)$ (top) and return ratio (bottom) as a function of the attacker budget.

The convolutional layers are replaced with fully-connected layers, and the final tanh activation is swapped for batch normalization with affine transformations.

**Evaluation.** We consider two performance metrics. The Kullback-Leibler (KL) divergence $D_{\mathrm{KL}}(S_v\|S_a)$ measures the discrepancy between the estimated state distribution $S_a$ and the victim's state distribution $S_v$, an aspect not previously quantified in model stealing. We represent $S_v$ with a reference normal distribution $\mathcal{N}(\boldsymbol{\mu}^*, (\boldsymbol{\sigma}^*)^2)$ estimated from a dataset of 1 million states, $\mathbb{S}_v$, collected from the interaction between the victim policy $\pi_v$ and the environment. The return ratio assesses the stealing performance by dividing the average return generated by the attacker policy in the environment by the average return of the victim policy. The return ratio is the average one derived from eight episodes with random initial state. We also employ five distinct random seeds to train five separate attacker policies and plot the result in Appendix E to account for any variability of Stealthy Imitation.

## 5.2 STEALTHY IMITATION ATTACK PERFORMANCE

We assess the effectiveness of various policy stealing methods, as shown in Figure 2. The measure of $D_{\mathrm{KL}}(S_v\|S_a)$ is specific to our approach (top row), as the Random strategy does not refine a distribution, and DFME focuses on fine tuning samples. We observe that the gap between $S_a$ and $S_v$ becomes consistently smaller and achieves convergence, even when starting from a high value in HalfCheetah. On average, we achieve an 81% reduction in $D_{\mathrm{KL}}(S_v\|S_a)$ across all environments. Our method substantially outperforms other attacks in terms of return ratio (Figure 2, bottom row). In the Hopper environment, we achieve a return ratio of 97% with just 5 million queries. In contrast, the best competing method, $\mathcal{N}(\mathbf{0}_n, \mathbf{10}_n^2)$, under the same query budget reaches only 70% and quickly falls below 25%. Further details on the performance of the reward discriminator can be found in Appendix F. While the Random $\mathcal{N}(\mathbf{0}_n, \mathbf{1}_n^2)$ baseline shows promise in the Hopper environment with 35 million queries, it does not maintain this performance as consistently as ours across varying query budgets. DFME does not manage to effectively steal the victim policy. This is largely due to its generator's tendency to search for adversarial samples within a predefined initial range, limiting its ability to explore other regions as flexibly as Stealthy Imitation.

## 5.3 ANALYSIS

**Distribution approximation.** In our approach, we employ a Gaussian of parameters $\boldsymbol{\mu}$ and $\boldsymbol{\sigma}$ to estimate the state distribution $S_a$. We assume that, as long as the distribution is close to $S_v$, the attacker can successfully steal the victim policy. To confirm this hypothesis, we train $\pi_a$ via behavioral cloning for 200 epochs on five different distributions for $S_a$, each approximated directly from the real state dataset $\mathbb{S}_v$: (i) and (ii) $\mathcal{N}(\boldsymbol{\mu}^*, (\boldsymbol{\sigma}^*)^2)$ and $\mathcal{N}(\boldsymbol{\mu}^*, \boldsymbol{\Sigma}^*)$: the mean $\boldsymbol{\mu}^*$ and variance $(\boldsymbol{\sigma}^*)^2$ or covariance $\boldsymbol{\Sigma}^*$ are directly calculated from $\mathbb{S}_v$, representing diagonal and full covariance matrix, respectively; (iii) and (iv) $\hat{S}_{v,u}$ and $\hat{S}_{v,m}$: these are non-parametric distribution approximations derived using kernel density estimation (KDE), treating variables as independent and dependent, respectively; and (v) $\mathbb{S}_v$: This samples data directly from the real states. Figure 3 shows that successful policy stealing is feasible even when queries are sampled from an approximate distribution.

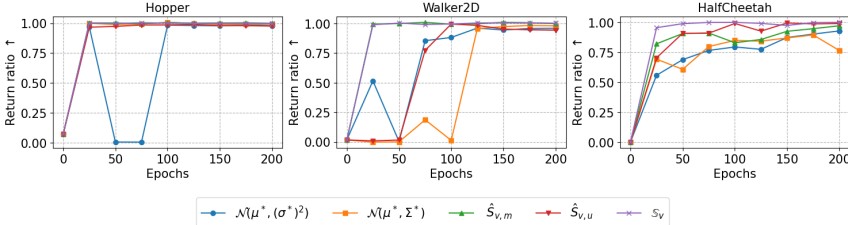

Figure 3: Model stealing success for different choices of $S_a$ based on the underlying distribution $S_v$.

Table 1: Spearman's rank correlation between validation loss $\bar{\mathcal{L}}_b$ and distribution divergence $D_{\mathrm{KL}}$.

| Task | $(\bar{\mathcal{L}}_b, D_{\mathrm{KL}})$ | |
| --- | --- | --- |
| | **Correlation $\rho$** | **p-value** |
| Hopper | $-0.84$ | $4.79 \times 10^{-164}$ |
| Walker2D | $-0.78$ | $7.59 \times 10^{-122}$ |
| HalfCheetah | $-0.81$ | $4.01 \times 10^{-140}$ |

The normal distribution with diagonal covariance matrix is an appropriate choice due to its low number of parameters. Moreover, Appendix D experimentally shows that our method is robust to estimation errors on both $\boldsymbol{\mu}$ and $\boldsymbol{\sigma}$.

**Correlation between difficulty of imitation and distribution divergence.** To empirically evaluate the hypothesis that the difficulty of imitation is correlated with the divergence between $S_a$ and $S_v$, we create 600 estimated state distributions $S_a$. These distributions are parameterized as $S_a(\boldsymbol{s}; \boldsymbol{z}\boldsymbol{\sigma}^* + \boldsymbol{\mu}^*, \boldsymbol{\sigma}^*)$, where each element of $\boldsymbol{z}$ is randomly sampled from a uniform distribution over $[0, 4]$, and its sign is chosen randomly. As a result, the KL divergences $D_{\mathrm{KL}}(S_v \| S_a)$ for these estimated state distributions range approximately from 0 to 8. For each $S_a$, we construct a transfer dataset of $10^5$ points and train the attacker's policy $\pi_a$ using BC for one epoch. We measure the average validation loss $\bar{\mathcal{L}}_b$ as a proxy for the difficulty of imitation. We apply Spearman's rank correlation test to these measurements, and the results are summarized in Table 1. These results demonstrate a statistically significant correlation for $(\bar{\mathcal{L}}_b, D_{\mathrm{KL}})$, thus supporting the use of $\pi_e$ as a reliable distribution evaluator in Section 4.3.

**Ablative analysis.** We study the impact of each component of our method by systematically removing them one at a time, while keeping the other components unchanged. The ablation study includes: (i) the use of $b_v \times \bar{\boldsymbol{\sigma}}$ instead of $b_v$ samples of transfer dataset $\mathcal{D}_v$ to train the distribution evaluator $\pi_e$; (ii) bypassing the **reward model training** and directly using the validation loss $\mathcal{L}_b$ of each sample as weight for the **reward-guided distribution refinement**; (iii) skipping the pruning step of the transfer dataset before training the reward model; and (iv) using $b_v$ instead of $b_v \times \bar{\boldsymbol{\sigma}}$ to train attacker's policy $\pi_a$ during **behavioral cloning**. The result is depicted in Figure 4. We observe that incorporating a reward model can more efficiently minimize the distribution divergence $D_{\mathrm{KL}}(S_v \| S_a)$. Additionally, employing a fixed budget for the evaluator model helps the attacker select a better $S_a$, thereby improving the return ratio. We also note the stabilizing effect of pruning the transfer dataset prior to training the reward model. Moreover, if a dynamic budget is not used when constructing the transfer dataset, we observe undesired shifts in $S_a$ over iterations in Hopper and this leads to a significant reduction in the return ratio.

## 5.4 STEALTHY IMITATION ATTACK COUNTERMEASURE

In this experiments, we test the efficiency of the proposed defense to our Stealthy Imitation attack. To this end, we consider the input range to match the minimum and maximum values encountered during training. Upon detecting a query that is outside the predefined input range, the victim policy will uniformly sample an action as a response. The results in Figure 5 indicate that the countermeasure substantially impedes the attacker's ability to approximate the victim's distribution, consequently reducing the return ratio of the attacker policy.

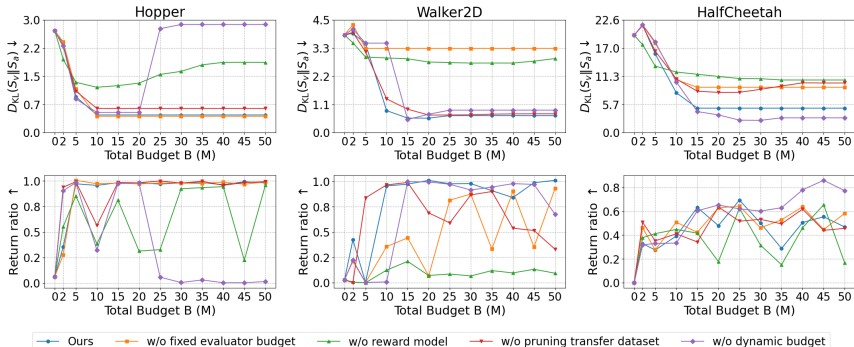

Figure 4: Ablation study of SI attack. We validate the necessity of (i) fixing the dataset size to train the evaluator model, (ii) dynamic budget, (iii) reward model, and (iv) pruning the transfer dataset.

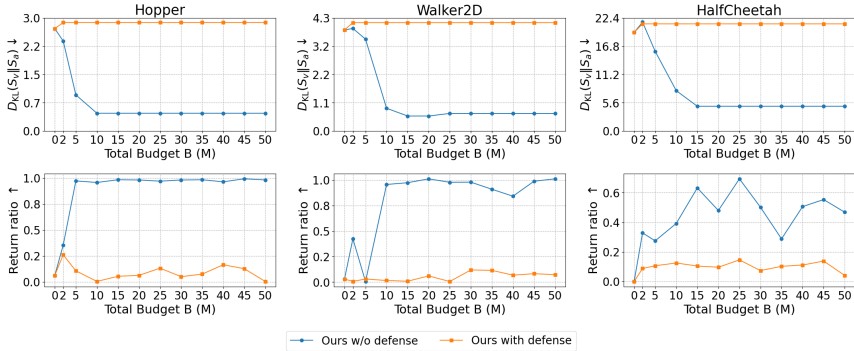

Figure 5: Efficiency of the proposed defense against Stealthy Imitation.

## 6 DISCUSSION

**Computational efficiency.** In addition to theft effectiveness, Stealthy Imitation also demonstrates computational efficiency. The main computational load comes from training the attacker policy $\pi_a$ on the optimized transfer dataset $\hat{\mathcal{D}}$. This is more computational efficient compared to utilizing all data with size of total budget $B$ like random strategy.

**Limitations and future work.** The limitations in this work present opportunities for future research and exploration. Firstly, attackers should consider the potential effects of initial distribution discrepancies. While our method, initializing the estimated distribution with a standard Gaussian, has proven effective, the threshold beyond which initial distribution divergence compromises effectiveness remains to be identified. Secondly, Stealthy Imitation, being agnostic to the victim RL algorithm, can adapt to various victim policies trained with other RL strategies; however, performance may vary across different RL algorithms and requires further examination. Finally, expanding our approach to other domains, where acquiring the input range is challenging such as those involving feature vectors and multivariate time series predictions, holds considerable promise for future research.

## 7 CONCLUSION

We show for the first time that an attacker can successfully steal policies in control systems without requiring environment access or prior knowledge of the input range—a strong attack vector that has not been demonstrated or considered in prior research. Lacking access to the victim data distribution, we show that a Gaussian assumption for the attacker query data is sufficient for efficient policy extraction. Our Stealthy Imitation attack outperforms existing methods adapted to policy stealing for a limited-knowledge attacker. We show that it is harder to imitate the victim policy when the distribution of the attack queries increasingly aligns that of the victim, thus allowing an attacker to refine their query distribution. We encourage policy owners to consider the risks of stealing and to use available defenses, such as the one proposed in this paper, to protect their assets.

## REPRODUCIBILITY STATEMENT

The authors are committed to ensuring the reproducibility of this work. The appendix provides extensive implementation details, and the code and setup will be made publicly available as open-source.

## ETHICS STATEMENT

While we are demonstrating an attack in this work, it is not targeted to a specific system but rather a generic attack vector. Therefore, in best practice no responsible disclosure procedure is necessesary or would even be possible. It is in fact of high importance to make developers and deployment aware of such risks, and thus such type of attacks are commonly published in AI/ML, and particular S&P venues. Additionally, we directly propose a defense and encourage policy owners to use it. The authors strictly comply with the ICLR Code of Ethics[1].

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

## A  ALGORITHMS

We now provide a detailed description of each function in Algorithm 1, along with their pseudo code.

**Query action.**  Query action (Algorithm 2) is the function where we obtain the transfer dataset from victim policy and attacker policy. We sample $b$ state vectors from a Gaussian distribution parameterized by $\boldsymbol{\mu}$ and $\boldsymbol{\sigma}$, and obtain responses $\boldsymbol{a}$ from the policy $\pi$. When $\pi = \pi_v$, the output is dataset $\mathcal{D} = \mathcal{D}_v$; otherwise, it is $\mathcal{D}_a$ when policy is $\pi_a$.

---
**Algorithm 2** QueryAction

---
**Require:** Policy $\pi$, mean $\boldsymbol{\mu}$ and standard deviation $\boldsymbol{\sigma}$, query budget $b$,
**Ensure:** Dataset $D$
1: Sample $b$ data points $\boldsymbol{s}$ from $\mathcal{N}(\boldsymbol{\mu}, \boldsymbol{\sigma}^2)$
2: $\boldsymbol{a} \leftarrow \pi(\boldsymbol{s})$
3: $\mathcal{D} := \{(\boldsymbol{s}_i, \boldsymbol{a}_i) | i = 1, ..., b\}$
4: **return** $\mathcal{D}$

---

**Behavioral cloning.**  We train policy $\pi$ to mimic the state-action pair mapping in dataset $\mathcal{D}$ via supervised learning by minimizing the Huber loss, i.e., behavioral cloning in Algorithm 3. Considering that the attacker policy $\pi_a$ has different dataset size requirement as distribution evaluator $\pi_e$ using behavioral cloning, we use an additional demand size $N$ to control it.

---
**Algorithm 3** BehavioralCloning

---
**Require:** Dataset $\mathcal{D} = \{(\boldsymbol{s}_i, \boldsymbol{a}_i^*)\}$, policy $\pi$, demand size $N$, epochs $E$, learning rate $\eta$
**Ensure:** Updated policy $\pi$
1: Sample $N$ data from $\mathcal{D}$ and split into training and validation $\mathcal{D}_t$ and $\mathcal{D}_v$
2: **for** $e = 1$ to $E$ **do**
3:     **for** each batch $(\boldsymbol{s}, \boldsymbol{a}^*)$ in $\mathcal{D}_t$ **do**
4:         Calculate loss $\mathcal{L}_b \leftarrow \text{HuberLoss}(\pi(\boldsymbol{s}), \boldsymbol{a}^*)$         ▷ Compute loss using Huber loss
5:         $\pi \leftarrow \pi - \eta \nabla_\pi \mathcal{L}_b$         ▷ Update model parameters using gradient descent
6:     **end for**
7: **end for**
8: **return** $\pi$

---

**Train reward.**  We use the code pipeline provided in engine Contributors (2021) to train the reward model in Algorithm 4, except for the additional function PruneData. Reward model is trained for total 400 steps in each iteration with learning rate $\eta = 10^{-3}$.

---
**Algorithm 4** TrainReward

---
**Require:** dataset $\mathcal{D}_a$ queried from attacker policy, dataset $\mathcal{D}_v$ queried from victim policy, reward model $\hat{\mathcal{R}}$, demand size $N$, total steps $T$, learning rate $\eta$
**Ensure:** Trained reward model $\hat{\mathcal{R}}$
1: Sample $N$ data from $\mathcal{D}_v$ and split into training and validation $\mathcal{D}_{vt}$ and $\mathcal{D}_{vv}$
2: $\mathcal{D}'_{vt} \leftarrow \text{PruneData}(\mathcal{D}_{vt})$
3: **for** $i = 1$ to $T$ **do**
4:     Sample batch data $(\boldsymbol{s}_v, \boldsymbol{a}_v)$ from $\mathcal{D}'_{vt}$ and $(\boldsymbol{s}_a, \boldsymbol{a}_a)$ from $\mathcal{D}_a$
5:     $L_v \leftarrow -\log(1 - \hat{\mathcal{R}}(\boldsymbol{s}_v, \boldsymbol{a}_v))$
6:     $L_a \leftarrow -\log(\hat{\mathcal{R}}(\boldsymbol{s}_a, \boldsymbol{a}_a))$
7:     $\nabla L \leftarrow \nabla(L_v + L_a)$         ▷ Compute the gradient of the total loss
8:     $\hat{\mathcal{R}} \leftarrow \hat{\mathcal{R}} - \eta \nabla L$         ▷ Update the reward model
9: **end for**
10: **return** The trained reward model $\hat{\mathcal{R}}$

---

**Prune data.**  When the action is equal to maximum or minimal value, i.e., extreme action, it is less likely to be the normal action predicted by the victim policy on the real state distribution, as most control systems do not prefer such extreme action. Extreme action value can easily cause

instability in control systems. By pruning the transfer dataset shown in Algorithm 5, the reward model can identity the difference of state-action pairs coming from the victim and attacker policies. For instance, if there is a state-action pair whose action is an extreme value, then the reward model tends to identity it as a state-action pair from the attacker, as there is no such data in the transfer dataset querying the victim policy after pruning.

---

**Algorithm 5** PruneData

---

**Require:** Dataset $\mathcal{D}$
**Ensure:** Cleaned Dataset $\mathcal{D}'$
1: $\mathcal{D}' \leftarrow \emptyset$
2: **for** each $(\boldsymbol{s}_i, \boldsymbol{a}_i)$ in $\mathcal{D}$ **do**
3:      **if** no element of $\boldsymbol{a}_i$ equals 1 or $-1$ **then**
4:          $\mathcal{D}' \leftarrow \mathcal{D}' \cup \{(\boldsymbol{s}_i, \boldsymbol{a}_i)\}$
5:      **end if**
6: **end for**
7: **return** $\mathcal{D}'$

---

**Distribution evaluate.** The function described in Algorithm 6 is exactly the same as behavioral cloning, but the final output of the function is the validation loss $\bar{\mathcal{L}}_b$ of evaluator $\pi_e$.

---

**Algorithm 6** DistributionEvaluate

---

**Require:** Dataset $\mathcal{D} = \{(\boldsymbol{s}_i, \boldsymbol{a}_i^*)\}$, policy $\pi$, portion size $N$, epochs $E$, learning rate $\eta$
**Ensure:** validation loss $\bar{\mathcal{L}}_b$
1: Sample $N$ data from $\mathcal{D}$ and split into training and validation $\mathcal{D}_t$ and $\mathcal{D}_v$
2: **for** $e = 1$ to $E$ **do**
3:      **for** each batch $(\boldsymbol{s}, \boldsymbol{a}^*)$ in $\mathcal{D}_t$ **do**
4:          Calculate loss $\mathcal{L}_b \leftarrow \text{HuberLoss}(\pi(\boldsymbol{s}), \boldsymbol{a}^*)$          $\triangleright$ Compute loss using Huber loss
5:          $\pi \leftarrow \pi - \eta \nabla_\pi \mathcal{L}_b$          $\triangleright$ Update model parameters using gradient descent
6:      **end for**
7: **end for**
8: Calculate average validation loss $\bar{\mathcal{L}}_b$ on $\mathcal{D}_v$
9: **return** $\bar{\mathcal{L}}_b$

---

**Distribution refinement.** We apply Equation (7) on the validation split of the transfer dataset to calculate the new $\boldsymbol{\mu}$ and $\boldsymbol{\sigma}$, described in Algorithm 7.

---

**Algorithm 7** DistRefine

---

**Require:** dataset $\mathcal{D}$, reward model $\hat{\mathcal{R}}$, demand size $N$
**Ensure:** updated $\boldsymbol{\mu}'$ and $\boldsymbol{\sigma}'$
1: Sample $N$ data from $\mathcal{D}$ and split into training and validation $\mathcal{D}_t$ and $\mathcal{D}_v$
2: $\mathcal{D}'_v \leftarrow \text{PruneData}(\mathcal{D}_v)$
3: $\boldsymbol{\mu}' \leftarrow \frac{\sum_{(s,a) \in \mathcal{D}'_v} \hat{r}(s,a) \cdot s}{\sum_{(s,a) \in \mathcal{D}'_v} \hat{r}(s,a)}$          $\triangleright \hat{r}(\boldsymbol{s}, \boldsymbol{a}) = -\log(\hat{\mathcal{R}}(\boldsymbol{s}, \boldsymbol{a}))$
4: $\boldsymbol{\sigma}'^2 \leftarrow \frac{\sum_{(s,a) \in \mathcal{D}'_v} \hat{r}(s,a) \cdot (s - \mu')^2}{\sum_{(s,a) \in \mathcal{D}'_v} \hat{r}(s,a)}.$
5: $\boldsymbol{\sigma}' = \sqrt{\boldsymbol{\sigma}'^2}$
6: **return** $\boldsymbol{\mu}'$ and $\boldsymbol{\sigma}'$

---

# B    ENVIRONMENT AND VICTIM POLICY

We conducted our experiments on environments sourced from Gymnasium (Towers et al., 2023). The specific environments, along with their version numbers and the performance metrics of the victim policies, are detailed in Table 2. The victim policies are trained using the Ding repository (engine Contributors, 2021), a reputable source for PyTorch-based RL implementations (Paszke et al., 2017). We employ SAC to train the victim policy; hence, the victim policy comprises an actor and a critic model. The actor model receives the state as input and outputs the action distribution, while the critic model receives a concatenated state and action as input and outputs the Q-value. During

Table 2: Environments and performance of victim policy.

| Environment | Observation space | Action space | Victim return |
|---|---|---|---|
| Hopper-v3 | 11 | 3 | 3593±3 |
| Walker2D-v3 | 17 | 6 | 4680±43 |
| HalfCheetah-v3 | 17 | 6 | 12035±61 |

queries to the victim policy, only the actor model is utilized, outputting the mean of the action distribution as a response. The state observations primarily consist of the positional coordinates and velocities of various body parts.

## C  INFLUENCE OF MODEL ARCHITECTURE

We investigate the impact of various attacker policy architectures on performance when executing Stealthy Imitation. Each victim policy utilizes a three-layer fully-connected network with 256 hidden units. To understand the effect of architecture variations, we modify the attacker policies by adjusting the layer numbers to 4, 6, and 10. Furthermore, we conduct experiments with the original layer structure, but reduce the hidden units to 128.

We depict the results on Figure 6. To better understand the impact, except for $D_{\text{KL}}(S_v \| S_a)$ and return ratio, we also provide raw $D_{\text{KL}}(S_v \| S_a)$ on top row, which is the last $D_{\text{KL}}(S_v \| S_a)$ at the end of the iteration, rather than the one selected by distribution evaluator $\pi_e$. We observe that the raw $D_{\text{KL}}$ of different architecture choices exhibit similar tendencies, thus the architecture choice has limited impact on the distribution refinement. In the second row of Figure 6, except for Walker2d, the selection of the $D_{\text{KL}}(S_v \| S_a)$ during refining by $\pi_e$ guarantee an appropriate estimated distribution $S_a$ and low $D_{\text{KL}}(S_v \| S_a)$, preventing the divergence of distribution approximation. However, we observe that the return ratio exhibits higher variance in the third row. This indicates that the return ratio is sensitive when the architecture is different, even when the estimated distribution is closed to the real state distribution. This is also a challenge in the realm of imitation learning, known as compounding errors (Syed & Schapire, 2010; Ross et al., 2011; Xu et al., 2020). Compounding errors imply that even minor training errors can snowball into larger decision errors. In our case, the minor training error comes from different architecture choices.

It is essential to highlight that this issue of compounding errors is predominantly absent in image classification model stealing, where test data points are independently evaluated. Nonetheless, the robustness of the estimation of the underlying distribution $S_v$ in terms of KL divergence underscores the effectiveness of our approach.

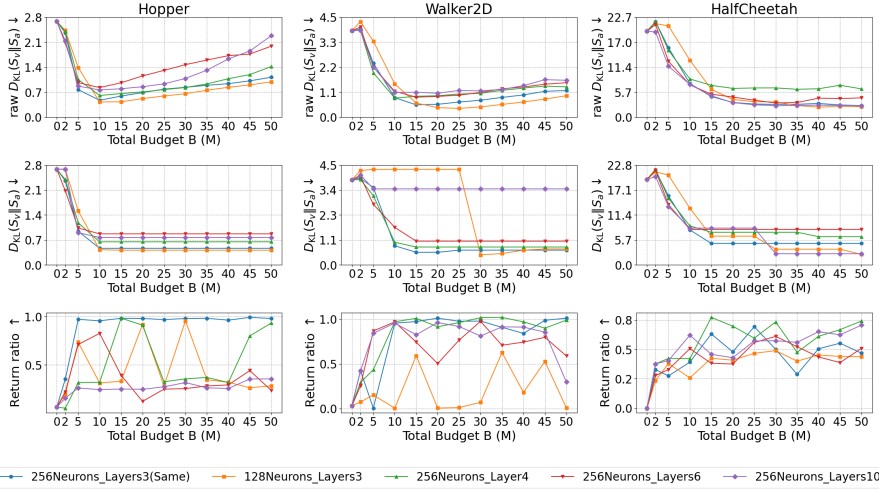

Figure 6: Influence of model architecture on stealing performance.

# D    ROBUSTNESS TO DISTRIBUTION APPROXIMATION ERRORS

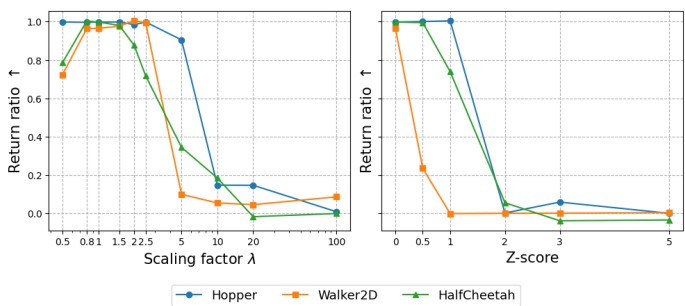

Figure 7: Left: policy stealing performance (return ratio) when $\boldsymbol{\mu} = \boldsymbol{\mu}^*$ and the scale factor $\lambda$ modifies $\boldsymbol{\sigma}^*$ such that $S_a = \mathcal{N}(\boldsymbol{\mu}^*, (\lambda\boldsymbol{\sigma}^*)^2)$. Right: policy stealing performance (return ratio) with $\boldsymbol{\sigma} = \boldsymbol{\sigma}^*$ and $\boldsymbol{\mu} = \boldsymbol{z}\boldsymbol{\sigma}^* + \boldsymbol{\mu}^*$, such that $S_a = \mathcal{N}(\boldsymbol{z}\boldsymbol{\sigma}^* + \boldsymbol{\mu}^*, (\boldsymbol{\sigma}^*)^2)$.

We customize $S_a$ with different parameters to explore the effect of discrepancy between $S_a$ and $S_v$. The left of Figure 7 explores the impact of varying $\boldsymbol{\sigma}$ while holding $\boldsymbol{\mu} = \boldsymbol{\mu}^*$ constant such that $S_a = \mathcal{N}(\boldsymbol{\mu}^*, (\lambda\boldsymbol{\sigma}^*)^2)$ with a factor $\lambda$. Conversely, the right investigates the effect of modifying $\boldsymbol{\mu}$ while keeping $\boldsymbol{\sigma} = \boldsymbol{\sigma}^*$ constant, $S_a = \mathcal{N}(\boldsymbol{z}\boldsymbol{\sigma}^* + \boldsymbol{\mu}^*, (\boldsymbol{\sigma}^*)^2)$. Different values of $\boldsymbol{z}$ serve as a measure of the divergence between the estimated $\boldsymbol{\mu}$ and $\boldsymbol{\mu}^*$. The sign of each element in $\boldsymbol{z}$ is randomly chosen. Transfer datasets, each containing 1 million queries, are generated from these customized distributions. These datasets are then used to train the attacker's policy $\pi_a$ through BC for up to 2000 epochs, utilizing early stopping with a patience of 20 epochs. From Figure 7 we observe that minor variations in $\boldsymbol{\sigma}$ are more tolerable compared to deviations in $\boldsymbol{\mu}$.

# E    VARIABILITY OF STEALTHY IMITATION

We report the variability of Stealthy Imitation in Figure 8 by using five random seeds to obtain five estimated distributions $S_a$ and train five attacker policies $\pi_a$. The performance of each policy is still obtained by collecting the average return ratio from eight episodes. We observe that the variability of $D_{\text{KL}}(S_v \| S_a)$ has impact on that of the return ratio, suggesting that a reliable estimated distribution is crucial to attacker policy training.

# F    PERFORMANCE OF THE REWARD DISCRIMINATOR

In this section, we analyze how the reward discriminator loss defined in Equation (6) changes throughout the distribution estimation process (Figure 9). In each iteration, we train the reward model for 400 steps; in each step, a batch of data will be sampled from the current victim and attacker distributions, $\mathcal{D}_v$ and $\mathcal{D}_a$ respectively. The x axis in Figure 9 represents the number of steps using a total of 50 million query budget.

We observe that the reward discriminator exhibits oscillations with the variation of the estimated distribution and attacker policy through the iterations. The discriminator's loss may decrease when

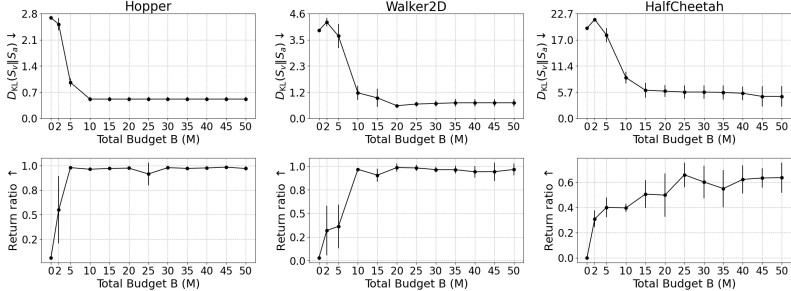

Figure 8: Variability of policy stealing performances.

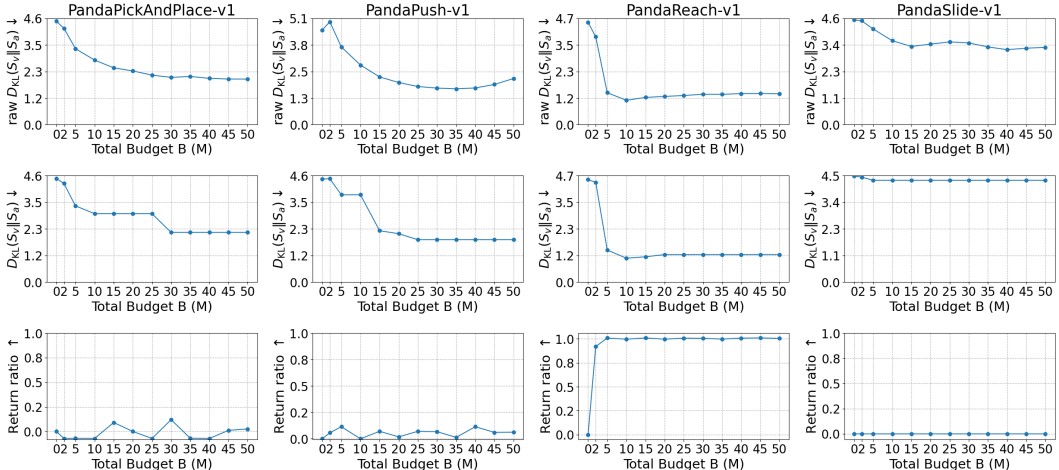

Figure 9: The reward discriminator loss in Equation (6).

it successfully identifies attacker's state and action pair data but can increase again as the estimated distribution shifts to a new region where the reward model has not been trained.

## G  FRANKA EMIKA PANDA ROBOT POLICY STEALING

We validate Stealthy Imitation in a realistic scenario where the victim policies are trained for a Franka Emika Panda robot arm provided by panda-gym (Gallouédec et al., 2021). All victim policies are from official repository in HuggingFace[2], and we focus on four out of five tasks, excluding the stacking task where the victim policy exhibited a 0% success rate. In the four tasks we select, the maximum and minimum returns are 0 and -50, respectively.

**Experimental setup.** To demonstrate efficacy, we initialize the estimated distribution with $\mathcal{N}(\boldsymbol{\mu}^* + 3\boldsymbol{\sigma}^*, (\boldsymbol{\sigma}^*)^2)$, with all initial $D_{\mathrm{KL}}$ being 4.5. Initializing with $\mathcal{N}(\mathbf{0}_n, \mathbf{1}_n^2)$ already results in a very small $D_{\mathrm{KL}}$ below 4.5. The attacker's architecture consists of a six-layer fully-connected network (512 hidden units, ReLU activation). The training involves five epochs for the attacker policy per iteration, with other hyperparameters mirroring those in the Mujoco setup. We calculate the return ratio using $\frac{R(\tau_a)-R(\tau_r)}{R(\tau_v)-R(\tau_r)}$, where $R(\tau_r)$ is the return of a randomly initialized attacker policy.

**Stealthy Imitation results.** Table 3 summarizes the task details and the percentage reduction in $D_{\mathrm{KL}}$ using SI after 50M queries. For full results, see Figure 10. SI effectively reduces $D_{\mathrm{KL}}$ in most tasks, except Slide due to the lower victim return. Higher victim returns correlate with greater $D_{\mathrm{KL}}$ reductions. The attacker's return ratio reaches 100% in the Reach task, but is near 0% in others, suggesting that precise distribution estimation is crucial for successful policy stealing.

Figure 10: Panda: raw distribution estimation measured by $D_{\mathrm{KL}}$ (top); evaluator $\pi_e$'s selected final estimated distribution (middle); return ratio of the attacker policy (bottom)

---

[2]https://huggingface.co/sb3

Table 3: Panda environments and reduction of $D_{\mathrm{KL}}$.

| Environment | Observation space | Action space | Victim return | $D_{\mathrm{KL}}$ reduction |
|---|---|---|---|---|
| PandaPickAndPlace-v1 | 26 | 4 | -12±12 | 52% |
| PandaPush-v1 | 25 | 3 | -6±2 | 59% |
| PandaReach-v1 | 13 | 3 | -2±1 | 73% |
| PandaSlide-1 | 25 | 3 | -29±13 | 4% |

