# OpenReview forum: "Stealthy Imitation: Reward-guided Environment-free Policy Stealing"
_ICLR.cc/2024/Conference — Submitted to ICLR 2024_

### Official Review · Reviewer_Maon · 2023-10-31

**Soundness:** 3 good
**Presentation:** 3 good
**Contribution:** 2 fair
**Rating:** 6
**Confidence:** 3

**Summary:**

The paper proposes an attack for stealing deep neural network policies without access to the environment or input states' ranges. The paper also proposes a defense for countering this attack. The attack includes creating a dataset by querying the victim policy, BC for training the attacker, a discriminative reward to distinguish state-action pairs from attacker or from policy, reward weighted sampling of states in the next iteration of querying the victim policy.

**Strengths:**

1. Interesting problem formulation. The challenges beyond supervised model stealing are clear.
2. Strong performance of the proposed obfuscation defense to the proposed attack.

**Weaknesses:**

1. Policy stealing is performed on simplistic mujoco case studies with simple MLPs. While the threat model is realistic, the application seems simple. Moreover, there are no experiments with realistic robotic imitation learning datasets (e.g. robomimic, D4RL's Adroit, CARLA) and recent models (e.g. diffusion policies, behavior transformers). This is particularly important given that model stealing is particularly relevant with large models only available as API such as the recent RT-2-X model [1]. The authors provide an abltaion with number of layers in an MLP but not with other architectures.

[1] Padalkar, Abhishek, et al. "Open X-Embodiment: Robotic learning datasets and RT-X models." 2023.

2. The performance of the learned reward discriminator and the performance of the victim policy+defense is not shown.
2. The code has not yet been made available. While the paper promises to release the code, it is limiting to not examine the code during the review process.

**Questions:**

1. What would the required budget be to scale to image or lidar inputs?
2. Is there perhaps a way to break the defense along the lines of adaptive attacks proposed in [2]?

[2] Tramer et al. "On Adaptive Attacks to Adversarial Example Defenses." 2020.

---

> ### Author Response · Authors · 2023-11-16
> **Response to Reviewer Maon**
>
> We appreciate the thoughtful and beneficial feedback provided by the reviewer. We are pleased that reviewer find the challenges beyond supervised model stealing clear. We would like to address the following questions:
>
> > 1. While the threat model is realistic, the application seems simple.
>
> We add new experiments to validate Stealthy Imitation in realistic scenarios where the victim policies are trained for a Franka Emika Panda robot arm provided by panda-gym [1]. Please see our [general comment about Validation of Applicability in Realistic Scenarios](https://openreview.net/forum?id=7q7s5fXEpP&noteId=ebtZUbFh9r) for a summary. We have also updated the paper draft to include the experiment details on the Panda tasks, which you can find in Appendix G.
>
>
> > 2. This is particularly important given that model stealing is particularly relevant with large models only available as API such as the recent RT-2-X model.
>
> We want to clarify that image models such as RT-2-X do not fit the setup of the knowledge-limited attacker we consider.
> Particularly, the input range is known for the image domain (i.e., pixel values in [0,255]). For this case, classic existing model stealing methods can be applied.
>
>
> > 3. The performance of the victim policy+defense is not shown.
>
> The performance of the victim policy with defense in a normal operating environment remains unchanged compared to without defense. This is because the defender only outputs incorrect actions when the queried state falls outside the valid input range.
>
> > 4. The performance of the learned reward discriminator is not shown.
>
> The loss of the reward discriminator exhibits oscillations with the variation of the estimated distribution and attacker policy through the iterations (similar to GAN pipelines).
> We have updated the paper draft with the reward discriminator's loss in Appendix F.
>
> > 5. What would the required budget be to scale to image or lidar inputs?
>
> As clarified previously, the assumptions of an unknown input range to the attacker does not hold for images or lidar. As such, this extension is not within the scope of the paper.
>
> > 6. Is there perhaps a way to break the defense along the lines of adaptive attacks proposed in (Tramer et al. "On Adaptive Attacks to Adversarial Example Defenses." 2020.)?
>
> We would like to emphasize that [Tramer et al., 2020] and other results in that line of work address adversarial examples, a different type of security threat than model stealing; as such, these are not directly applicable to our setup.
>
> Nevertheless, we can consider the consequences of the attacker becoming aware of the defense strategy.
> In that case, our defense can be enhanced to produce responses with similar variance for queries outside the range as those within range.
> For more details, please also see our [answer to Reviewer 3FCu's 4th point](https://openreview.net/forum?id=7q7s5fXEpP&noteId=XiSDuxa5Ma) on possible attack variations.
>
> Could the reviewer let us know if these variations answer the scenarios they had in mind?
>
>
> [1] Quentin Gallouédec, Nicolas Cazin, Emmanuel Dellandréa, and Liming Chen. panda-gym: Open-Source Goal-Conditioned Environments for Robotic Learning. Robot Learning Workshop: Self-Supervised and Lifelong Learning at NeurIPS, 2021

---

> ### Comment · Reviewer_Maon · 2023-11-22
>
> The authors have addressed my concerns. I am raising my score. But the absence of code is still particularly unhelpful.

---

### Official Review · Reviewer_AqHM · 2023-11-03

**Soundness:** 2 fair
**Presentation:** 2 fair
**Contribution:** 2 fair
**Rating:** 6
**Confidence:** 3

**Summary:**

This paper considers the potential for model stealing attacks on control policies trained with deep reinforcement learning. While model stealing has been applied to image classifiers in supervised learning settings, the authors argue that the control setting is more difficult because the distribution of state inputs may be unknown. The authors formulate a method to estimate the state distribution via the hypothesis that the policy should be more difficult to imitate within this distribution, an assumption which is verified in Table 1. The method involves jointly estimating the state distribution and learning an imitation policy over that state distribution with behavior cloning. The authors show it is more accurately able to both estimate the state distribution and steal the victim policy compared to using a fixed state distribution or using DFME.

**Strengths:**

The experiments in the paper seem to be relatively comprehensive, including an ablation study, validation of the assumptions underlying the authors' algorithm, and a defense against the attack. The results also seem quite promising, showing that the SI attack estimates both the state distribution and victim policy well. I don't know of previous work on model stealing in deep RL/control, so the work seems novel, although I am not very familiar with the area.

**Weaknesses:**

Some potential weaknesses include:
 * The writing could be clearer in some places. The proposed algorithm has many components, and some of the experiments are somewhat complex—it was a bit hard to understand the purpose of some algorithm components or experiments at first.
 * The setting of wanting to steal a policy without knowing the environment seems unrealistic—what is the attacker planning to do with the policy if it doesn't have access to the environment? Wouldn't the point of stealing a policy be to run the policy in the environment, which entails the attacker has access?
 * The defense shows that a very simple countermeasure, which is trivial to implement and has no impact on normal system performance, prevents this attack from succeeding. This could also be viewed as a positive, though, since calling attention to this type of attack and associated defense could drive practitioners to employ the defense in real-world applications. Maybe it would be good to reframe the contribution of the defense along these lines.
 * There are some additional issues with some of the exposition being unclear or misleading:
    * Section 4.2 part I: the authors write "we choose a univariate normal distribution" but in fact it is a multivariate normal distribution since the state space is $\mathbb{R}^n$. I believe the authors mean that they use a multivariate normal distribution with a diagonal covariance matrix.
    * The "smooth L1 loss" referred to in section 4.2 part II is generally known as the Huber loss and the authors should cite Huber, "Robust Estimation of a Location Parameter" (1964).
    * It feels a bit misleading that the y-axes in the top row of Figure 2 don't start at 0. It looks at a glance like the KL approaches zero with more queries in each column, whereas in fact all of them seem to asymptote above 0.
 * The comparison to DFME might not be fair—see my questions below.

**Questions:**

Based on the weaknesses above, here are some questions for the authors. If these are satisfactorily answered, I can raise my score:
 * Why is the objective in (1) to minimize reward difference? Shouldn't it either be to maximize reward, or imitate as precisely as possible?
 * What is the point of stealing a policy if you don't know the environment?
 * What exactly is the purpose of the experiments in Figure 3, since to calculate all of these distributions one needs access to states sampled from $S_v$, which is not possible under the threat model proposed? Why not compare to $\\mathcal{N}(\\mathbf{\\tilde\\mu}, \\mathbf{\\tilde\\sigma}^2)$ as well?
 * Regarding the comparison to DFME, the authors write that "DFME does not manage to effectively steal the victim policy. This is largely due to its generator’s tendency to search for adversarial samples within a predefined initial range." What exactly is the initial state range output by DFME, and what happens if you expand the initial range? For instance, one could simply use a larger weight initialization for the final layer of the generator network in DFME, or multiply the outputs by a large constant. How does this compare to the newly proposed SI attack?

---

> ### Author Response · Authors · 2023-11-16
> **Response to Reviewer AqHM**
>
> We thank the reviewer for their constructive comments and valuable suggestions. It is encouraging that the reviewer finds the paper novel and comprehensive. We answer their questions as follows.
>
> > 1. There are some additional issues with some of the exposition being unclear or misleading:[...]
>
> We appreciate the suggestions for improving the clarity of the paper and have incorporated them into our revised draft.
>
> > 2. Why is the objective in (1) to minimize reward difference? Shouldn't it either be to maximize reward, or imitate as precisely as possible?
>
> Minimizing reward difference focuses on the target of replicating the functionality of the victim policy, as opposed to obtaining the best possible performance on the task. This goal of functional stealing is standard throughout model stealing literature [1].
> Additionally, it is common in inverse RL [2] to learn a reward function that measures the actions in the context of the overall goal, rather than scoring the replicating of individual actions. For instance, consider the following pathological case: if an attacker policy imitates only the second half of a victim policy's actions in a sequence, the action agreement is 50\%. However, differing actions in the first half could result in zero returns for the attacker policy due to cascading effects, highlighting the significance of targeting the end goal, rather than exact action replication.
>
>
> > 3. What is the point of stealing a policy if you don't know the environment?
>
> We consider two reasons why an attacker might seek to steal a policy without knowledge of the environment:
>
> Firstly, **further attacks**: stealing a policy enables the attacker to disrupt the target control system. They can generate adversarial signals with the stolen policy, even without understanding the environment.
>
> Secondly, **sensitive information exposure and unauthorized usage**: it enables the attacker to access confidential information, like sensor types or preprocessing methods. The information could potentially aid in the development of their own control system.
>
>
> > 4. What exactly is the purpose of the experiments in Figure 3, since to calculate all of these distributions one needs access to states sampled from $\mathbb{S}_v$, which is not possible under the threat model proposed? Why not compare to $\mathcal{N}(\tilde{\mu},\tilde{\sigma}^2)$ as well?
>
> This experiment is conducted from an analytical viewpoint, rather than that of an attacker. This goes for the entire analysis and ablations section (Sec. 5.3). This experiment in particular explores the minimal knowledge required for successful policy stealing. It highlights the risk associated with exposing input distributions, even through a normal distribution, and demonstrates that RL-trained policies in control systems can be compromised using supervised learning. Since the attacker cannot access $\mathbb{S}_v$, Stealthy Imitation does not utilize it. The outcomes using the attacker's approximation, $\mathcal{N}(\tilde{\mu},\tilde{\sigma}^2)$, are displayed at the bottom of Figure 2, marked as "ours".
>
>
> > 5. What exactly is the initial state range output by DFME, and what happens if you expand the initial range? For instance, one could simply use a larger weight initialization for the final layer of the generator network in DFME, or multiply the outputs by a large constant. How does this compare to the newly proposed SI attack?
>
> We aligned DFME's initial state output distribution with ours $ \mathcal{N}(\mathbf{0}_n,\mathbf{1}_n^2)$. The original DFME generator used a tanh activation function, confining outputs to [-1,1], typical in image classification model stealing. We adapt DFME to the best of our ability to the present setup by substituting this with **batch normalization with learnable scaling and shift factors**, enabling the generator to explore regions beyond the $ \mathcal{N}(\mathbf{0}_n,\mathbf{1}_n^2)$ starting point.
>
> Additionally, we follow the reviewer's suggestion and amplify the output of the DFME generator by factors of 10 and 100. The resulting return ratio of the attacker policy in these two additional experiments is worse than without amplification throughout 50M queries.
>
>
> [1] Tribhuvanesh Orekondy, Bernt Schiele, and Mario Fritz. Knockoff nets: Stealing functionality of black-box models. In Conference on Computer Vision and Pattern Recognition (CVPR), pp. 4954–4963, 2019b.
>
> [2] Ng, Andrew Y., and Stuart Russell. Algorithms for inverse reinforcement learning. ICML. 2000.

---

> > ### Comment · Reviewer_AqHM · 2023-12-01
> > **Response to authors**
> >
> > I appreciate the authors' comprehensive rebuttal. I think the authors have generally addressed my concerns, so I have raised my score.
> >
> > It would be good to more clearly emphasize in the paper the reasons that an attacker would want to steal a policy when they don't have access to the environment. It would also be good to include the additional experiments with DFME.

---

### Official Review · Reviewer_3FCu · 2023-11-05

**Soundness:** 2 fair
**Presentation:** 4 excellent
**Contribution:** 3 good
**Rating:** 5
**Confidence:** 4

**Summary:**

The authors present a new approach for black-box policy stealing: reproducing a policy without access to the environment/data that it was originally trained on.  The authors only assume a fixed budget of interaction with the model being attacked (the victim) and do not assume differentiability.  They also do not assume knowledge of the input range to the model.  The authors' method attempts to find an approximation of the input range and data used to train the model by fitting a normal distribution to input values with a high prediction mismatch between current behavior predicted by the attacker and the victim.  They then train the attacker by sampling from this distribution and using the victim's predictions as labels.  The authors also present a defense against their method of attack that selects random responses for inputs outside of the acceptable input range.

**Strengths:**

* The idea is new and poses interesting technical challenges.
* The paper is also clear, well structured and well explained.
* The ablations are well thought out and provide a lot of insight into the details of the technique.

**Weaknesses:**

* The test environments are quite limited.  These Mujoco environments are small, and there are only three of them.  The only contact dynamics are with the ground and self-collision.  Testing in larger environments with more degrees of freedom and richer dynamics is highly encouraged.
* This method seems impossible for policies with high-dimensional input such as images.
* The approximation of the state distribution using a normal distribution seems quite limiting, and it's an open question as to whether this would work for more complex high-dimensional problems.  The authors discuss this in 5.3 and provide some empirical experiments purporting to show that this is fine, but I am somewhat skeptical about the situation in problems with larger state spaces.
* The primary method proposed here seems like it would be prone to distribution shift over time.  If I understand correctly, during each iteration, $\pi_a$ is trained on $D_v$ which consists of only on the most recently collected data.  At the same time $\mu$ and $\sigma$ are estimated by approximating the distribution of points where $\pi_a$ struggles to match the output of the victim $\pi_v$.  It seems likely in this scenario that because the distribution that $\pi_a$ is trained on shifts over time (as $\mu$ and $\sigma$ change), it will get better at the regions of space most recently encountered, and may lose capability on regions that have not been visited recently.  It seems that the use of $\pi_e$ which is reinitialized every time is meant to address this, but some discussion of the role of drift would be interesting here.
* It also seems that there are a lot of cat-and-mouse games that could be played with the proposed defense against this stealing technique as well.  The authors suggest randomizing outputs for states that are outside of a known input range, but this is trivially circumvented by querying the same point multiple times and assuming points with low variance are within the training distribution (a lot of this depends on whether the policy is known to be stochastic).  If the defender then tries to keep out-of-bounds points the same or close across multiple queries, then the attacker could start querying very close points.  The defender could then try to build some random, yet fixed, yet smoothly changing landscape of points outside the boundaries, but this seems like a complicated modelling task.  In the end, the suggestion of returning random points in the out-of-bounds range seems right, but like most security challenges, hard to get right in a way that prevents further circumvention.
* The authors target interactive policies in RL domains, but there doesn't seem to be anything here requiring RL, or specific to the RL setting.  This technique should be applicable in any domain where you are trying to clone a network with input small enough that it can be modeled using a normal distribution.  With that in mind, it would be good to see results on a wider variety of problems.

**Questions:**

* Addressing the questions about drift in the Weaknesses section would be great.
* Discussion the issues with the counter-attack would also be beneficial.

Although my score is currently marginally below the acceptance threshold, I like this paper, and am quite flexible in my evaluation of it.  If the authors are able to demonstrate their method in a wider variety of challenging environments, this would go a long way to improving my score.

---

> ### Author Response · Authors · 2023-11-16
> **Response to Reviewer 3FCu**
>
> We thank the reviewer for the insightful and positive feedback. We are pleased to see that the reviewer found the idea and challenges new and potentially applicable across different domains. We answer their questions below.
>
> > 1. The test environments are quite limited. These Mujoco environments are small, and there are only three of them.
>
>
> We add new experiments to validate Stealthy Imitation in realistic scenarios where the victim policies are trained for a Franka Emika Panda robot arm provided by panda-gym [1]. Please see our [general comment about Validation of Applicability in Realistic Scenarios](https://openreview.net/forum?id=7q7s5fXEpP&noteId=ebtZUbFh9r) for a summary. We have also updated the paper draft to include the experiment details on the Panda tasks, which you can find in Appendix G
>
> > 2. This method seems impossible for policies with high-dimensional input such as images.
>
> We want to clarify that image data does not fit the setup of the knowledge-limited attacker we consider.
> Particularly, the input range is known for the image domain (i.e., pixel values in [0,255]). For this case, classic existing model stealing methods can be applied.
>
>
> > 3. The approximation of the state distribution using a normal distribution seems quite limiting
>
> We argue that the capacity to estimate states based on a normal distribution is a feature of our method.
> We show that this assumption is sufficient to approximate more complex distributions; this finding is part of the paper's contributions.
> The experimental results on the Panda robot arm show that this assumption holds for high dimensional state spaces (26 dimensions for the Panda Pick And Place task).
>
>
> > 4. [...] circumvent defense by querying multiple times
>
> We agree with the reviewer that tackling security topics often involves a cat and mouse game between the attacker and the defender.
> Building on the reviewer's attack suggestion, our defense can be enhanced to produce responses with similar variance for queries outside the range as those within range. For instance, the defender could map queries outside the range to a random or fixed point within the range, resulting in a similar output variance. We appreciate this discussion and will incorporate it into the revised draft of our paper.
>
> > 5. [...] it will get better at the regions of space most recently encountered, and may lose capability on regions that have not been visited recently.
>
> We believe the reviewer's understanding is correct. We clarify that the distribution shift across iterations is indeed intentional and a key part of our method in search of the correct victim distribution. The evaluator $\pi_e$ plays a crucial role in selecting the most suitable distribution. The final attacker policy only needs to perform well on the distribution chosen by $\pi_e$, not on all explored distributions.
>
>
> > 6. The authors target interactive policies in RL domains, but there doesn't seem to be anything here requiring RL, or specific to the RL setting.
>
> We thank the reviewer for highlighting the general potential of our method. We include here results for Stealthy Imitation applied to binary classification on the Diabetes dataset [2]. The input consists of eight variables and has different semantics and scales; the output can be considered as a continuous "action" between [0,1]. We initialize the estimated distribution with $\mathcal{N}(\mu^*+3\sigma^*,(\sigma^*)^2)$, instead of $ \mathcal{N}(\mathbf{0},\mathbf{1})$ which results in high $D_{\mathrm{KL}}$ (2712). The accuracy of the victim model is 77\%. The final attacker model can achieve 69\% accuracy (90\% w.r.t. the victim) and the $D_{\mathrm{KL}}$ also has a decreasing trend as shown below.
>
> | Iteration         | 0   | 2   | 4   | 6   | 8   | 10  | 12  |
> |-------------------|-----|-----|-----|-----|-----|-----|-----|
> | $D_{\mathrm{KL}}$ | 4.5 | 4.4 | 4.0 | 3.6 | 3.2 | 3.1 | 3.1 |
>
> While we acknowledge the potential of Stealthy Imitation beyond RL contexts (e.g., feature vectors, multivariate time series), we leave the in-depth analysis of these applications to future work.
>
> [1] Quentin Gallouédec, Nicolas Cazin, Emmanuel Dellandréa, and Liming Chen. panda-gym: Open-Source Goal-Conditioned Environments for Robotic Learning. Robot Learning Workshop: Self-Supervised and Lifelong Learning at NeurIPS, 2021
>
> [2] Kahn, Michael. Diabetes. UCI Machine Learning Repository. https://doi.org/10.24432/C5T59G.

---

> > ### Comment · Reviewer_3FCu · 2023-11-22
> > **Panda Environment**
> >
> > 1 and 3: The fact that the return ratio is near zero for all but (what seems to be) the simplest Panda task seems to demonstrate the weaknesses of this approach rather than it's benefits.  While improving DKL says something, it's not what we really care about, which is the success of the policy rolled out in the actual environment.  To me this points to the need for better distribution modelling than simply using a normal distribution with diagonal covariance matrix.
> >
> > 5. Right, you definitely want to see a different distribution each time, but my point is that because you are training $\pi_a$ (which then influences $\hat{\mathcal{R}}$) on new data exclusively at each step, it seems like there's nothing preventing you from oscillating back and forth between different data modes over several rounds of the outer loop.  Training on the new data alone means you may catastrophically forget the old data from previous rounds, which will then make it more appealing to go back and explore those regions again because they are no longer modeled well.  I think the normal fix to this is to continue training $\pi_a$ on data from previous rounds as well in order to prevent the catastrophic forgetting.
> >
> > Unfortunately I think I need to keep my score at 5.  I think there are some really interesting ideas here, but I think it would be better served by exploring other choices/approaches of distribution modelling, and finding ways to actually approximate the policies of some of these more challenging environments in a way that improves the return ratio.

---

### Official Review · Reviewer_LEDd · 2023-11-10

**Soundness:** 3 good
**Presentation:** 3 good
**Contribution:** 2 fair
**Rating:** 5
**Confidence:** 3

**Summary:**

This paper introduces a method for stealing reinforcement learning policies without access to the training environment. This approach, called 'Stealthy Imitation,' efficiently estimates the state distribution of the target policy and utilizes a reward model to refine the attack strategy. While showcasing superior performance in policy stealing compared to existing methods, the paper also presents a practical countermeasure to mitigate such attacks. However, the complexity of the approach and ethical considerations surrounding its potential misuse are notable concerns.

**Strengths:**

- Innovative Approach: Represents an advancement in the field of model stealing, particularly in RL settings, with a more realistic threat model.

- Effective Methodology: Demonstrates superior performance in stealing RL policies compared to existing methods, even with limited information.

- Practical Countermeasure: Offers a realistic and practical solution to mitigate the risks of such attacks.

**Weaknesses:**

- Complexity: The method's complexity, particularly in estimating the state distribution and refining the attack policy, is relatively straightforward and not incurring significant novelty/advancements compared to the existing approaches.

- Real-World Applicability: Transitioning from a controlled experimental setup to real-world applications might present unforeseen challenges, for example, the authors only experiment on simple tasks that have relatively simple state distribution. The underlying challenges for applying such tasks to a more complicated task is unclear.

**Questions:**

The authors propose a new method to advancing the model stealing attack against RL models and the reviewer appreciates that the authors also provide a countermeasure method for the ethical concerns. However, the reviewer has major concerns regarding the novelty of the methods. Model stealing with a synthetic dataset + a surrogate model (here state estimators) is not new to the community. The reviewer would appreciate if the authors could elaborate more on the challenges of this matter, especially the unique challenges when applying such attack in the RL setting. Also, the baseline of DFML is not fair enough given the different settings here. Following the philosophy of DFML, it seems that the proposed SI is the same approach under the RL setting. Another issue the reviewer has concerns is regarding the complexity of the target RL tasks. While it might be easy to build a synthetic dataset for simple tasks like Hopper, it might be infeasible for more complicated tasks as the state dimension increases. The authors should have a more thorough discussion regarding how such attacks could have impacts in the real world (e.g., stealing policies that are more valuable, which are often trained for handling more complicated tasks).

---

> ### Author Response · Authors · 2023-11-16
> **Response to Reviewer LEDd**
>
> We thank the reviewer for the thoughtful feedback. We are encouraged that the reviewer finds that policy stealing under the new threat model to be an advancement and that the countermeasure is practical. We aim to answer the reviewer's concerns in the following.
>
>
> > 1. Real-World Applicability: Transitioning from a controlled experimental setup to real-world applications might present unforeseen challenges
>
> We add new experiments to validate Stealthy Imitation in realistic scenarios where the victim policies are trained for a Franka Emika Panda robot arm provided by panda-gym [1]. Please see our [general comment about Validation of Applicability in Realistic Scenarios](https://openreview.net/forum?id=7q7s5fXEpP&noteId=ebtZUbFh9r) for a summary. We have also updated the paper draft to include the experiment details on the Panda tasks, which you can find in Appendix G.
>
> > 2. Model stealing with a synthetic dataset + a surrogate model (here state estimators) is not new to the community.
>
> We would like to emphasize that our method differs from standard model stealing by focusing on how to estimate the valid input range, while we do follow the standard way to steal policy after the estimation process. Additionally, if by "state estimators" here the reviewer refers to the estimated probability distribution, then it is also different from previous works which train a generator model through back-propagation. We also include a reward model which helps understand the imitation difficulty across regions. For a detailed explanation on how our method contrasts with DFME, please also see the answer to the next point.
>
> > 3. [...] the baseline of DFML is not fair enough given the different settings here.
> > Following the philosophy of DFML, it seems that the proposed SI is the same approach under the RL setting.
>
> Stealthy Imitation is the first method to steal a policy without the knowledge of input range, which means that there are no baselines for this setup. However, we want to provide a comparison to prior art and adapt DFME to the best of our ability to the present setup.
> We also summarize the difference between DFME and Stealthy Imitation in the following table; the two methods tackle different threat models and achieve different goals.
>
>
> | Method         | Data Source              | Target Feature            | Optimization Focus                            | Reusable                                                                     | Key Advantage                                                 |
> |----------------|--------------------------|---------------------------|------------------------------------------------|---------------------------------------------------------------------------------|---------------------------------------------------------------|
> | DFME    | Model-generated data     | Adversarial examples      | Sample difficulty with $L_1$ loss | Not, as adversarial examples vary from model to model              | Refines data distribution using knowledge of input range |
> | Ours   | Probability distribution | Hard-to-Imitate Region   | Overall difficulty with reward model | Yes, as the input range does not change           | Efficiently estimates input range                 |
>
> > 4. challenges of this matter, especially the unique challenges when applying such attack in the RL setting
>
> The main challenge is the potentially unbounded input space and the valid range being unknown to the attacker. Current model stealing mainly focuses on the image domain, where the input range is known (i.e., pixel range [0,255]). This premise does not hold in, e.g., industrial control systems, where variables have varying semantics and scales (e.g., sensor data). For these systems, the model is often trained using RL instead of supervised learning.
> We show that such RL models can also be stolen using supervised learning on synthetic data (Sec. 5.3, Distribution approximation), as long as the attacker knows the input distribution.
> In this setup, the difficulty thus becomes the input distribution estimation, which we solve as part of our contribution.
>
> [1] Quentin Gallouédec, Nicolas Cazin, Emmanuel Dellandréa, and Liming Chen. panda-gym: Open-Source Goal-Conditioned Environments for Robotic Learning. Robot Learning Workshop: Self-Supervised and Lifelong Learning at NeurIPS, 2021

---

### Author Response · Authors · 2023-11-16
**Validation of Applicability in Realistic Scenarios**

We thank all the reviewers for their insightful feedback and want to answer one common concern here: the test environments are limited.

To answer the reviewer's concern, we include new experiments to validate Stealthy Imitation in realistic scenarios where the victim policies are trained for a Franka Emika Panda robot arm provided by panda-gym [1]. All victim policies are from official
repository in [HuggingFace](https://huggingface.co/sb3) and we focus on four out of five tasks, excluding the stacking task where the victim policy exhibited a 0\% success rate. In the four tasks we select, the maximum return is 0 and the minimum return is -50.


To demonstrate our method's efficacy, we initialize the estimated distribution with $\mathcal{N}(\mu^*+3\sigma^*,(\sigma^*)^2)$, thus all initial $D_{\mathrm{KL}}$ is 4.5, because initializing with $ \mathcal{N}(\mathbf{0},\mathbf{1})$ already results in a very small $D_{\mathrm{KL}}$ below 4.5.

The table below summarizes task details and the percentage reduction in $D_{\mathrm{KL}}$ using Stealthy Imitation after 50M queries. We observe that Stealthy Imitation effectively reduces $D_{\mathrm{KL}}$ in most tasks, except Slide due to the lower victim return. Higher victim returns correlate with greater $D_{\mathrm{KL}}$ reductions. The attacker's return ratio reaches 100\% in the Reach task but is near 0\% in others, suggesting that precise distribution estimation is crucial for successful policy stealing in certain tasks.

| **Environment**         | **Observation space** | **Action space** | **Victim return** | **$D_{\mathrm{KL}}$ reduction** |
|-------------------------|-----------------------|------------------|-------------------|---------------------------------|
| PandaPickAndPlace-v1    | 26                    | 4                | -12±12            | 52%                             |
| PandaPush-v1            | 25                    | 3                | -6±2              | 59%                             |
| PandaReach-v1           | 13                    | 3                | -2±1              | 73%                             |
| PandaSlide-1            | 25                    | 3                | -29±13            | 4%                              |

We have updated the paper draft with complete experiments on the Panda tasks in Appendix G.

[1] Quentin Gallouédec, Nicolas Cazin, Emmanuel Dellandréa, and Liming Chen. panda-gym: Open-Source Goal-Conditioned Environments for Robotic Learning. Robot Learning Workshop: Self-Supervised and Lifelong Learning at NeurIPS, 2021

---

### Meta-Review · Area_Chair_ae3p · 2023-12-06

**Metareview:**

(a) Summarize the scientific claims and findings of the paper:

The paper introduces "Stealthy Imitation," a method for black-box policy stealing: reproducing a policy without access to the environment/data that it was originally trained on. The authors only assume a fixed budget of interaction with the model being attacked (the victim) and do not assume differentiability nor knowledge of the input range to the model. The method finds an approximation of the input range and data used to train the model by fitting a Gaussian distribution to input values with a high prediction mismatch between current behavior predicted by the attacker and the victim. This is then used to train the attacker by sampling from the distribution and using the victim's predictions as labels. The paper compares against other policy stealing baselines across a range of environments.

(b) Strengths of the paper:
(+) Clear Presentation (Reviewer 3FCu): The paper is well structured, clear, and provides insightful ablations.
(+) Effective Methodology and Ablations (Reviewer LEDd, AqHM, 3FCu): Demonstrates superior performance in stealing RL policies compared to existing methods. The ablations are insightful.
(+) Practical Countermeasure (Reviewer LEDd): Offers a realistic and practical solution to mitigate the risks of such attacks.

(c) Weaknesses of the paper:
(-)  Real-world applicability (Reviewer LEDd, Reviewer 3FCu):  Distribution modeling is challenging in high-dimensional state spaces, and the choices made here seem limiting. While the idea is interesting, the approach boils down to trying to discover the input distribution of a model that you have black-box access to. To solve the input distribution question, the authors use a very simple diagonal Gaussian, which means you are just trying to find the approximate range of each dimension. This is unlikely to work for challenging high-dimensional problems: complex multi-joint systems often have dynamics that lead to coupling behaviors that could only be modeled with more complex distributions.

(-) Contrived setup (AqHM, 3FCu): The attack scenarios seem rather contrived and impractical. Various assumptions about getting unlimited access to a model but not the input range need to be justified.

(-) Limited Scope in Test Environments (Reviewer 3FCu): The test environments, primarily Mujoco and Panda robotic tasks, are too limited and simplistic.

**Justification For Why Not Higher Score:**

While the idea is interesting, the choice of distribution modeling seems simplistic, and the attack scenario seems contrived.

**Justification For Why Not Lower Score:**

N/A

---

### Decision · Program_Chairs · 2024-01-16

Reject